# L-Rhamnose Globally Changes the Transcriptome of Planktonic and Biofilm *Escherichia coli* Cells and Modulates Biofilm Growth

**DOI:** 10.3390/microorganisms12091911

**Published:** 2024-09-19

**Authors:** Charlotte E. Hantus, Isabella J. Moppel, Jenna K. Frizzell, Anna E. Francis, Kyogo Nagashima, Lisa M. Ryno

**Affiliations:** Department of Chemistry and Biochemistry, Oberlin College, Oberlin, OH 44074, USA

**Keywords:** biofilm, carbohydrate metabolism, L-rhamnose, RNA-seq, extracellular polymeric substances

## Abstract

L-rhamnose, a naturally abundant sugar, plays diverse biological roles in bacteria, influencing biofilm formation and pathogenesis. This study investigates the global impact of L-rhamnose on the transcriptome and biofilm formation of PHL628 *E. coli* under various experimental conditions. We compared growth in planktonic and biofilm states in rich (LB) and minimal (M9) media at 28 °C and 37 °C, with varying concentrations of L-rhamnose or D-glucose as a control. Our results reveal that L-rhamnose significantly affects growth kinetics and biofilm formation, particularly reducing biofilm growth in rich media at 37 °C. Transcriptomic analysis through RNA-seq showed that L-rhamnose modulates gene expression differently depending on the temperature and media conditions, promoting a planktonic state by upregulating genes involved in rhamnose transport and metabolism and downregulating genes related to adhesion and biofilm formation. These findings highlight the nuanced role of L-rhamnose in bacterial adaptation and survival, providing insight into potential applications in controlling biofilm-associated infections and industrial biofilm management.

## 1. Introduction

L-rhamnose is an abundant natural sugar found in plants and bacteria, and it has diverse biological functions. It is a primary component of both hemicellulose, a branched heteropolymer that lends structure to plant cell walls, and the *O*-antigen of lipopolysaccharides displayed on the outer membrane of pathogenic gram-negative bacteria [1,2]. Lectins on human skin cells recognize rhamnose displayed on the outer membrane, where it subsequently modulates calcium ion homeostasis and depresses the expression of genes related to growth, proliferation, and maintenance of the extracellular matrix [3]. Moreover, extracellularly presented rhamnose is implicated in increased bacterial invasion and pathogenesis in mammalian cell and organismal models [4,5]. Rhamnose is also the primary carbohydrate component of rhamnolipids: glycolipid biosurfactants that are predominantly produced by *Pseudomonas aeruginosa* and are notable in their ability to modulate biofilm formation, maturation, and dispersal [6,7,8,9]. Notably, groups have developed rhamnolipid mimics as a novel class of anti-biofilm compounds [10,11].

While *Escherichia coli*, a gram-negative bacterium, has an intrinsic rhamnose biosynthetic pathway, it also imports and catabolizes abundant environmental rhamnose. L-rhamnose enters the central metabolism via breakdown into the glycolytic/gluconeogenic intermediate dihydroxyacetone phosphate (DHAP) and S-lactaldehyde. The latter metabolite can either be oxidized to pyruvate or reduced by the S-1,2-propanediol oxidoreductase FucO to S-1,2-propanediol, an important starting material for the synthesis of many industrially relevant polymers and other chemicals. Initially, the partitioning of S-lactaldehyde to pyruvate or S-1,2-propanediol was thought to be driven by aerobic or anaerobic culture conditions, respectively. However, recent evidence suggests that L-rhamnose can be broken down into S-1,2-propanediol in aerobic conditions [12,13].

Genes coding for the enzymes required for rhamnose import (*rhaT*, a proton symporter), metabolism (*rhaBAD* operon), and regulation (*rhaRS* operon) are found at different loci in the K-12 genome [14,15]. The *rhaRS* operon is induced by the binding of L-rhamnose to RhaR, a transcription factor constitutively expressed at low levels. When rhamnose is present, RhaS, a member of the AraC/XylS family of transcriptional regulators, binds to L-rhamnose and engages the promoter region of the *rhaBAD* operon, activating transcription of those genes and therefore leading to degradation of L-rhamnose and tapering of induction [16]. The rhamnose-inducible promoter region ahead of the *rhaBAD* operon has been utilized extensively as an exogenous expression system that both responds linearly to the concentration of rhamnose in many different organisms and expresses heterologous proteins at high levels [17,18,19,20]. Currently, there are no other known promoter regions that RhaS or RhaR bind to when in the presence of L-rhamnose in *E. coli*, nor are there any other reported signaling effects of L-rhamnose in *E. coli*.

In the promoter regions of all the rhamnose-associated operons, there are also binding sites for the DNA-binding transcriptional dual regulator CRP, a transcription factor that prevents the expression of genes for nonpreferred carbon sources when glucose is present, thereby promoting the most efficient catabolic pathways [21,22,23]. Preferential carbon utilization influences 5–10% of all gene expression through a process called carbon catabolite repression, which is governed by CRP in the presence of cyclic-AMP and sRNA Spot 42, and which favors the use of glucose over other sugars [24,25,26]. The complex hierarchical utilization system of carbohydrates in *E. coli* has been well-studied [27,28,29,30]. L-rhamnose exists towards the bottom of this hierarchy in complex mixtures of sugars, only preferred to D-ribose when sugars are present in saturating (0.5% (*w*/*w*)) conditions [27].

L-rhamnose’s environmental abundance, use as an energy source, and role as a component of signaling molecules in biofilm formation and host-pathogen interactions elevates the importance of understanding its global impact on cellular processes. While there have been many transcriptomic and proteomic studies that examine changing gene expression and translation in *E. coli* in response to various carbon sources (e.g., Luria-Bertani culture medium [31], or certain sugars like glucose or lactose [32]), we were unable to find any information on how L-rhamnose might impact the global transcriptome. Other labs have explored the relationship between glucose availability in the media and repression of *E. coli* biofilms, and have explored the phenotypic effects of some sugars—but not rhamnose—on *E. coli* [33,34,35,36,37,38]. The effect of rhamnose on biofilm formation and composition has been documented for other gram-negative organisms like *Flavobacterium columnare* and *Phaeobacter inhibens*, but, to our knowledge, not in *E. coli* [39,40].

We were interested in comparing the influence of rhamnose on the cell under several experimental conditions, including (1) cells grown in planktonic culture as opposed to biofilm; (2) as the sole carbon source in minimal media or as an additional carbon source in rich media; and (3) in cooler (28 °C) as opposed to warmer (37 °C) growth temperatures. We chose to examine these two temperatures since our study strain, PHL628 *E. coli*, forms biofilms abundantly and most robustly at growth temperatures below 30 °C due to a point mutation in the OmpR protein that causes constitutive activation of the transcription factor CsgD [41,42,43]. CsgD regulates the expression of the proteins that assemble to form curli, which is an extracellular structure that aids in adhesion and biofilm formation. We also examined the presence of sub-saturating (0.1% *w*/*w*) vs. saturating (0.5% *w*/*w*) concentrations of rhamnose, as we and others have noted repression of biofilm formation with certain sugars at sub-saturating concentrations in rich media [44]. Here, we compare rhamnose’s impact on planktonic growth, biofilm formation, and the composition of extracellular polymeric substances (EPS) across all these conditions and present a comprehensive transcriptomic analysis, revealing global changes to gene expression with the presence of rhamnose.

## 2. Materials and Methods

### 2.1. Bacterial Strain Selection and Cultivation

Kanamycin (Sigma-Aldrich, St. Louis, MO, USA) was used to select for the PHL628 *E. coli* cells (a kind gift from Anthony G. Hay, Ph.D., Cornell University) for all experiments. Stock solutions of 10% (*w*/*w*) L-rhamnose (Sigma-Aldrich) and 10% (*w*/*w*) D-glucose (Sigma-Aldrich) were made by dissolving 1.00 g of L-(+)-rhamnose or D-glucose in 9.00 mL of ultrapure water and sterile filtering the solution using a 0.2 µm polyethersulfone membrane syringe filter (VWR, Radnor, PA, USA). A stock solution of 50 mg/mL (1000X) kanamycin was prepared in ultrapure water and sterile filtered prior to use.

All *E. coli* cell cultures were grown in lysogeny broth (LB), composed of 25.00 g of LB Lennox (Hardy Diagnostics, Santa Maria, CA, USA) per liter of water, or 1X M9 media containing 11.28 g of M9 salts (disodium phosphate (6.8 g/L), monopotassium phosphate (3.0 g/L), sodium chloride (0.5 g/L), and ammonium chloride (1 g/L) (BD Biosciences, Franklin Lakes, NJ, USA), per liter of water. All the *E. coli* cell colonies were grown on LB plates composed of 25.00 g Lennox LB broth and 12.00 g agar powder (Alfa Aesar, Haverhill, MA, USA) per liter of water. Depending on the experiment and type of selection needed, kanamycin, rhamnose, and/or glucose were added to the LB media and LB plates. An overnight starter culture was made of 8 mL of LB, 8 µL of 50 mg/mL kanamycin, and one colony of bacteria selected from the prepared agar plates, combined in a 15 mL conical tube with the cap placed loosely using tape to ensure air flow, and placed in a 37 °C shaking incubator for cell growth to occur overnight.

### 2.2. Growth Curves

A growth time course was created by measuring the OD_600_ of several different conditions over 15 h. A total of 20 µL of starter culture was diluted into a new conical tube containing 5.5 mL of LB, and this solution was then returned to the 37 °C shaking incubator for 2 h to return the culture to an exponential growth phase.

Conditions for growth included either LB or 1X M9 media, 28 °C or 37 °C, and 0%, 0.1%, or 0.5% (*w*/*w*) of rhamnose or glucose. For each condition, 3 mL of solution was prepared. The 0% solutions contained 150 µL sterile water, 30 µL of exponential growth culture, 3 µL of 50 mg/mL kanamycin, and 2.817 mL of LB or 1X M9. The 0.1% samples contained 30 µL of 10% (*w*/*w*) sugar and 120 µL sterile water, and the 0.5% (*w*/*w*) samples contained 150 µL of 10% (*w*/*w*) sugar in place of the 150 µL sterile water in the 0% solutions. In a 96-well clear flat-bottom plate (Corning Life Sciences, Corning, NY, USA), 300 µL of solution was delivered into each well in the column corresponding to a given condition, resulting in eight technical replicates. Using the kinetics mode on a Spectramax i3X (Molecular Devices, San Jose, CA, USA), the instrument was set to record the OD_600_ value every 15 min for 15 h with a 5-s shake before reading. The temperature was set to either 28 °C or 37 °C.

Growth curve data was plotted in GraphPad Prism 10, where the mean and standard error were calculated. A nonlinear, logistic growth fit of the data provided the rate constants and maximum absorbance values. Significant differences in the logistic growth data were determined using an ordinary one-way analysis of variance (ANOVA) with a post-hoc Tukey test.

### 2.3. Crystal Violet Assay

Biofilm was grown with 0.2 g of sterile glass wool (Sigma-Aldrich) in a clear 6-well plate (VWR, Radnor, PA, USA) [45]. For the 0% condition, each well contained 5 mL of LB, 150 µL of starter culture, 5 µL of 50 mg/mL kanamycin, and 250 µL of sterile autoclaved water. For the 0.5% (*w*/*w*) condition, each well contained 5 mL of LB, 150 µL of starter culture, 5 µL of 50 mg/mL kanamycin, and 250 µL of either 10% (*w*/*w*) L-rhamnose or D-glucose (control). For 0.1% (*w*/*w*), all conditions were identical except that 50 µL of 10% (*w*/*w*) L-rhamnose or D-glucose was added. The 6-well plates were grown at either 28 °C or 37 °C for 48 h, with a media change performed at 24 h. An OD_600_ of the supernatant was measured after 24 and 48 h. After 48 h of growth, the bacterial cultures in each well were discarded and the glass wool was rinsed three times with ultrapure water using a serological pipette. A 0.1% (*w*/*v*) crystal violet solution used to stain the microtiter plate biofilms was made by dissolving 0.0500 g of crystal violet powder (Sigma-Aldrich) in 50 mL of MilliQ ultrapure water [46]. After rinsing, 5 mL of crystal violet was added to each well and left to incubate at room temperature for 15 min. Once stained, each well was rinsed three times with a pH 7.0 sodium phosphate buffer (Sigma-Aldrich and Alfa Aesar). The plates were then left to dry with the lids ajar overnight in a static 37 °C incubator. A 30% acetic acid mixture used to solubilize the crystal violet was made by mixing 75 mL of acetic acid (Sigma-Aldrich) and 175 mL of ultrapure water. The following day, the crystal violet adhered to the glass wool was solubilized in 30% acetic acid. Stained glass wool was added to a 15 mL conical tube using sterilized forceps, along with 6 mL of 30% acetic acid to solubilize the crystal violet adhering to the glass wool. The conical tubes were vortexed and shaken at 200 rpm for 10 min. To measure the amount of crystal violet present in each sample, a 96-well clear plate was used to measure the absorbance of a 1:10 dilution. The OD_590_ was measured using a Spectramax i3X plate reader and the OD_590_/OD_600_ ratio was used to quantify biofilm growth. Growth was plotted in GraphPad Prism 10, where the mean and standard error were calculated. Significant differences were determined using a two-way analysis of variance (ANOVA) with a post-hoc Tukey test.

### 2.4. Biofilm Growth on Agar and EPS Harvesting

Biofilm grown for analysis of extracellular polymeric substances was grown statically on 10 cm LB-agar plates (VWR) supplemented with 50 µg/mL kanamycin, with or without 0.5% (*w*/*w*) rhamnose or glucose [47]. Using a sterile wooden inoculating pick, three to five colonies of PHL628 *E. coli* were collected from an agar plate and suspended in 2 mL of sterile saline (0.9% *w*/*v* NaCl) in a 15 mL conical tube. The solution was vortexed and adjusted to an OD_600_ of 0.08 to 0.1. A sterile cotton swab was dipped into the conical tube and excess liquid was removed. The swab was streaked over the surface of the 0% and 0.5% (*w*/*w*) L-rhamnose or D-glucose agar plates with even distribution next to a lit Bunsen burner and allowed to dry. Plates were transferred and incubated at 28 °C or 37 °C for 48 h. 

After the 48 h incubation period, a sterile 25 cm cell scraper was used to remove the layer of biofilm from the top of the agar, which was then transferred to the microcentrifuge tube using a pipette tip, and the wet mass was recorded. A volume of 750 µL of 1.5 M NaCl in ultrapure water was added and gently vortexed into a homogeneous slurry. The solution was incubated at room temperature for 5 min, and vortexed every minute. Samples were centrifuged for 10 min at 5000× *g*. For each condition, approximately 350 µL of supernatant was transferred into two new sterile microcentrifuge tubes to be stored at −20 °C. The remaining pellet was left in the centrifuge tube in the 37 °C oven to dry overnight. The dry mass of the harvested cells was recorded and used to normalize the total amount of biofilm.

To quantify the protein concentration in the EPS, a bicinchoninic acid (BCA) protein assay was performed (ThermoScientific). A stock solution of BSA at a concentration of 2 mg/mL was employed to generate a standard curve, spanning concentrations from 25 to 2000 μg/mL BSA in a 1.5 M NaCl solution, including a blank sample (0 μg/mL BSA). A volume of 25 µL of each standard and 1:10 dilutions of EPS (stock: 1.5 M NaCl) was pipetted into a clear 96-well plate (Corning). All EPS unknowns were performed in duplicate. A volume of 200 µL of the working reagent was added to all wells and mixed via pipetting. The plate was incubated at 37 °C for 30 min, followed by 10 min at room temperature. Absorbance was measured at 562 nm using the Molecular Devices Spectramax i3X microplate spectrophotometer. A standard curve was generated by plotting the blank-corrected absorbance values at 562 nm for each BSA standard against their respective concentrations (μg/mL) and was used to calculate protein concentration.

A phenol-sulfuric acid assay was conducted for carbohydrate quantification (DuBois 1956) [48]. A sample of 200 µL was added to 200 µL of 5% (*v*/*v*) phenol (Sigma Aldrich) in ultrapure water along with 1.0 mL of concentrated sulfuric acid (Flinn Scientific) in individual microfuge tubes; the tubes were mixed and incubated at room temperature for 10 min. Samples were then mixed well and placed in a 30 °C water bath for 20 min. Glucose standards were prepared simultaneously. A volume of 200 µL of the treated samples was pipetted into a clear 96-well plate (Corning), and the absorbance at 490 nm was measured on a Spectramax i3X plate reader (Molecular Devices).

Protein and carbohydrate concentrations were plotted relative to the dry biofilm mass of the collected sample in GraphPad Prism 10, where the mean and standard error were calculated. Significant differences were determined using a two-way analysis of variance (ANOVA) with a post-hoc Tukey test.

### 2.5. Confocal Microscopy

Biofilm cells were grown on glass wool in a manner identical to the method described above in the crystal violet assay. An OD_600_ of the supernatant was measured after 24 and 48 h. After 48 h, a small piece of glass wool approximately 2.5 cm long was removed for each condition and washed three times with MilliQ ultrapure water in a petri dish to remove planktonic cells. For the calcofluor white (Sigma-Aldrich) staining, the small piece of washed glass wool was placed on a slide, and one drop (approximately 20 µL) of 10% KOH (Sigma-Aldrich) followed by one drop of calcofluor white was added using a transfer pipette, as indicated by the manufacturer. For the FilmTracer SYPRO Ruby (Invitrogen) staining, the sample of glass wool was placed in a 24-well plate (VWR) with 3 mL of FilmTracer SYPRO Ruby stain and incubated in the dark for 30 min. The sample was then gently rinsed once with MilliQ water and placed on a slide. A cover slip was placed on top of each sample. Confocal laser scanning microscopy (CLSM) was employed to analyze the PHL628 biofilm cells and extracellular polymeric substances using a Zeiss LSM 880 inverted confocal microscope (Carl Zeiss, Jena, Germany) equipped with MA-PMT and GaAsP array detectors. Samples were viewed through a 10X objective. FilmTracer SYPRO Ruby-stained samples were imaged using excitation from a 458 nm argon laser, while calcofluor white-stained samples were imaged using excitation from a 405 nm diode array. Fluorescence intensity analysis was completed by imaging five randomly located, identically sized areas on a slide in the same z-plane of greatest fluorescent intensity. Images were analyzed in ImageJ by determining a common threshold of fluorescence using Otsu thresholding and subsequently determining the mean integrated density of fluorescent areas using the “analyze particles” function [49,50,51]. This analysis provided us with comprehensive information about the spread and size of stained biofilm in two dimensions, as well as fluorescence intensity. The integrated density weighted to particle size was plotted in GraphPad Prism 10, where the mean and standard error were calculated. Significant differences were determined using a two-way analysis of variance (ANOVA) with a post-hoc Tukey test.

### 2.6. RNA Harvesting

A pH 7.0, 0.2 M potassium phosphate (Alfa Aesar) solution and a pH 7.4, 10 mM Tris-HCL (G Biosciences, St. Louis, MO, USA) solution were prepared for harvesting biofilm from glass wool. Planktonic and biofilm bacterial cells were grown in 500 mL Erlenmeyer flasks with glass wool. Approximately 1 g of glass wool was autoclaved at 121 °C for 30 min in a 500 mL Erlenmeyer flask and loosely covered with aluminum foil. Into each flask, 100 mL LB, 103 µL 50 mg/mL kanamycin, 3 mL starter culture, and 5.4 mL autoclaved MilliQ water or 10% (*w*/*w*) L-rhamnose were added. Flasks were incubated in a 28 or 37 °C rotary shaking incubator at 200 rpm. After 24 h, the supernatant in the flask was removed using serological pipettes, replaced with fresh media identical to the previous day, and returned to the appropriate shaking incubator. The media change was necessary for bacterial survival by providing abundant nutrients available for continued growth and replication for the 48-h duration. 

Approximately 24 h after the media change and 48 h after the start of the experiment, planktonic cells were harvested from the liquid supernatant and biofilm was collected after detachment from the glass wool. A sample of swimming cells was collected in 15 mL conical tubes and the rest of the media was decanted from the Erlenmeyer flask. The glass wool was gently rinsed three times with 50 mL of 0.2 M potassium phosphate buffer to remove any residual planktonic cells. The glass wool was then transferred into a 250 mL autoclaved Erlenmeyer flask, using flame-sterilize tweezers, with approximately 8 g of autoclaved glass beads. A volume of 10 mL of Tris-HCl buffer was added, and flasks were shaken at 300 rpm in the rotary shaker for 10 min. After shaking, the solution was decanted into 15 mL conical tubes and the optical density at 600 nm (OD_600_) was measured using an Eppendorf BioPhotometer for planktonic and biofilm samples. The Genomics Agilent calculator was used to calculate the volume of each condition needed to harvest approximately 1.2 × 10^9^ cells/mL in sterile microcentrifuge tubes. After centrifuging for 10 min at 8000× *g*, the supernatant was decanted and cell samples were either placed in the −80 °C freezer or processed immediately for RNA purification.

Planktonic and biofilm cell samples were either removed from the −80 °C freezer on ice or immediately taken from harvesting to purify. A volume of 1 mL of RNAprotect Bacteria Agent (Qiagen, Germantown, MD, USA) was added to the cell pellets and vortexed for 5 s, followed by a 5-min incubation at room temperature. Microcentrifuge tubes were centrifuged for 10 min at 5000× *g*, followed by decantation of the supernatant. A TE buffer consisting of 30 mM Tris-HCl and 1 mM EDTA at pH 8.0 (Invitrogen) was combined with lysozyme (Sigma Aldrich) for a final concentration 15 mg/mL lysozyme. This was further combined with 20 mg/mL proteinase K (ThermoFisher) for a final concentration of 3 mg/mL. 115 µL of a TE/lysozyme/proteinase K mixture was added and vortexed for 10 s. Following this addition, the microcentrifuge tubes were incubated at room temperature for 10 min, vortexing for 10 s every two minutes. The Qiagen RNeasy Kit was used to purify the samples. A ThermoFisher NanoDrop One was used to measure the concentration and purity of samples.

### 2.7. RNA-seq Data Collection and Analysis

For RNA-seq, samples of RNA were harvested and purified as described above. At least 5 µg of RNA was added to GenTegra Tubes (GenTegra, LLC, Pleasanton, CA, USA) and diluted to a final volume of 50 µL with RNase-free water (Qiagen). Samples were frozen with liquid nitrogen and lyophilized to remove water. Samples were then stored at ambient temperature. Samples were analyzed by Mr.DNA (Molecular Research LP, Shallowater, TX, USA). The lyophilized total RNA was resuspended in 25 µL of nuclease free water. The concentration of the RNA was determined using the Qubit^®^ RNA Assay Kit (Life Technologies, Carlsbad, CA, USA). A total of 1 µg of RNA was used for rRNA removal using the Ribo-Zero Plus rRNA Depletion Kit (Illumina). rRNA depleted samples were quantified and used for library preparation using the KAPA mRNA HyperPrep Kits (Roche), following the manufacturer’s instructions. Following the library preparation, the final concentration of all of the libraries was measured using the Qubit dsDNA HS Assay Kit (Life Technologies), and the average library size was determined using the Agilent 2100 Bioanalyzer (Agilent Technologies). The libraries were then pooled in equimolar ratios of 0.6 nM, and sequenced paired-end for 300 cycles using the NovaSeq 6000 system (Illumina). The data discussed in this publication have been deposited in NCBI’s Gene Expression Omnibus [52] and are accessible through GEO Series accession number GSE274311 (https://www.ncbi.nlm.nih.gov/geo/query/acc.cgi?acc=GSE274311, accessed 15 September 2024).

To analyze the RNA sequencing data, a data workflow in R v4.3.2 was developed. Workflow and source code can be accessed from the Ryno Lab Github RNA-seq Repository (https://github.com/OCRynoLab/RhamnoseRNAseq, accessed 15 September 2024). Briefly, RNA-Seq data quality control, alignment, quantification, and statistical calculations were performed using the workflow, with the reference genome and annotation of *E. coli* strain MG1655 from NCBI (genome sequence GenBank ID U00096.3; assembly ID GCA_000005845.2). Adapter sequences were trimmed with scythe v0.994, and low-quality ends were trimmed with sickle v1.33 [53,54]. HISAT2 v2.2.2.1 was used to align reads to the reference genome [55]. Salmon v1.10.2 was used to quantify the expression of transcripts [56]. Using the tx2gene program in the Bioconductor R package, we created GFF files and compared them to the assembled and annotated *Escherichia coli* str. K-12 substr. MG1655 (Taxonomy ID: 511145, GCA_000005845). Features of interest were listed as “genes”, with each gene identified by the MG1655 locus tag. The resulting GFF file contained 4566 features (4240 protein coding, 147 pseudogene, 71 noncoding RNA, 22 rRNA, and 86 tRNA). Differentially expressed genes (DEGs) were identified by subjecting raw counts to the edgeR package in R v4.3.2 [57]. The Benjamini and Hochberg False Discovery Rate (FDR) criterion was used to compute *P_adj_* values [58]. An absolute value of log_2_ fold change > 2 (i.e., a four-fold difference in either direction) and a *P_adj_*  <  0.001 was used as the threshold for selecting DEGs. The limma package was used to conduct principal component analysis of the normalized counts [59]. All data were plotted in GraphPad Prism 10. Linear regression was conducted on comparative analyses and the fit was reported in the respective figures.

Gene ontology analyses were conducted on DEGs using the databases PANTHER and EcoCyc [60,61]. Enrichment testing was conducted in PANTHER using the over-representation tool [62]. All data was plotted in GraphPad Prism 10.

### 2.8. RT-PCR/qPCR

Prior to quantification of samples through qPCR, purified RNA must be converted into cDNA via reverse transcription (RT). RT-PCR reactions were performed to prepare cDNA from total cellular RNA using the QuantiTect Reverse Transcription Kit (Qiagen). The Rotor-Gene SYBR Green dye (Qiagen), cDNA, and appropriate primers purchased from Integrated DNA Technologies (Appendix A) were used for amplifications (45 cycles of 2 min at 95 °C, 10 s at 95 °C, 30 sec at 60 °C) in a Qiagen Rotor-Gene Q instrument. All transcripts were normalized to the housekeeping gene *rrsA*, which was selected due to other labs’ validation of the gene [63], and all measurements were performed in triplicate. Data were analyzed using the ΔΔCt method, and error is reported as SEM. Data were plotted in GraphPad Prism 10, where the mean and standard deviation were calculated. Significant differences were determined using a two-way analysis of variance (ANOVA) with a post-hoc Tukey test.

## 3. Results

### 3.1. PHL628 Growth Kinetics and Biofilm Growth with Rhamnose

We examined whether L-rhamnose influenced the growth profile of PHL628 *E. coli* over the course of 18 h at different concentrations, temperatures, and media conditions as compared to D-glucose (Figure 1). As expected, more substantive growth occurred overall in rich media (LB) compared to minimal media (M9) for both temperatures studied. Bacterial growth rate in rich media was significantly different depending on the sugar and concentration added.

In all experimental conditions, though most noticeably in minimal media, glucose significantly increased the rate of growth and the maximal growth compared to the control. Rhamnose had a modest or nonremarkable effect on enhancing growth in both rich and minimal media. We analyzed the data using a logistic growth model and obtained the rate constant for each condition (Table 1).

In minimal media, the bacterial population plateau occurred at very different cell densities depending on the type of sugar used; we see a projected maximal growth for 0.5% (*w*/*w*) glucose added that is 210% or 193% that of 0.5% (*w*/*w*) rhamnose at 28 and 37 °C, respectively. In contrast, there was no significant change in maximal growth when comparing rhamnose with the 0% control at either temperature. In summary, we observed that there is little difference in the rate of growth and total amount of growth in the presence of rhamnose versus the control (no added sugar), except for the 0.5% (*w*/*w*) rhamnose in minimal media at 37 °C. The addition of D-glucose to either rich or minimal growth media enhanced the growth rate and lead to greater maximum cell density (Table 1).

We considered whether the addition of rhamnose influenced the amount of PHL628 biofilm formation for our different experimental conditions by employing crystal violet staining of biofilm grown on glass wool. In rich media, rhamnose significantly depressed biofilm growth, particularly at warm growth temperatures (Figure 2A). For these experiments, the amount of biofilm formed, as measured by UV-visible absorbance of crystal violet at 595 nm, was normalized to the amount of planktonic (swimming) cells as determined by an OD_600_ measurement of the culture medium.

While we observed a modest decrease in biofilm formed with the addition of 0.1% (*w*/*w*) glucose in rich media at 37 °C, there was no significant difference in biofilm formation at higher concentrations of glucose at either temperature. In minimal media, we noted that in the absence of any added sugar, there was effectively no biofilm or planktonic cell growth and, therefore, we saw increased biofilm formation relative to the control with the addition of either sugar at both temperatures (Figure 2B). We observed only modest (<2-fold) significant increases in biofilm growth for those cultures grown at 28 °C for 0.5% (*w*/*w*) rhamnose and for both concentrations of glucose.

We and others have observed that biofilm growth in response to changing environmental conditions can be inherently dependent on substrate [64,65]. For many of our analysis techniques, we grew biofilm on glass wool in shaking flasks and therefore obtained information about cultures that can remain in the planktonic phase or partition to the biofilm/sedentary phase. We also grew biofilms on static agar plates that were supplemented with our varied experimental nutrients, finding that, in rich media, the addition of 0.5% (*w*/*w*) rhamnose significantly increased the amount of biofilm mass, independent of culture temperature, whereas glucose addition did not (Appendix A). Interestingly, we observed that when rhamnose is the only carbon source present in agar growth experiments, the amount of biofilm formed is significantly less than that observed with glucose as the only carbon source, mirroring the planktonic growth experiments shown in Figure 1. We attributed the differences we observed in rhamnose’s influence on biofilm growth to substrate type and indicate in the following sections which substrate was used to collect our data.

### 3.2. Influence of Rhamnose on EPS Protein and Carbohydrate Concentrations

We examined the effect of the addition of rhamnose on the composition of the extracellular matrix by specifically focusing on the protein and carbohydrate components. We found that the total protein concentration of extracellular polymeric substances (EPS) in our biofilms grown on rich agar media for 48 h was depressed compared to the 0% control, significantly so at 37 °C; adding glucose resulted in no significant change at 28 °C and increased protein concentration at 37 °C (Figure 3A). In minimal media, we observed significant increases in protein concentration for biofilms with added rhamnose compared to the 0.5% (*w*/*w*) glucose control at both growth temperatures (Figure 3B), though there was no significant difference in growth between the two sugars. We also quantified protein in the EPS using confocal microscopy and the dye SyproRuby, which stains extracellular protein (Appendix A).

For confocal experiments, we grew biofilm on glass wool; as this is particularly challenging at 37 °C, we thus focused on 28 °C growth. Interestingly, our SyproRuby confocal experiments in rich media did not wholly agree with our protein concentration experiments for biofilm grown on agar, highlighting the importance of substrate in EPS composition and the need for normalization to total biofilm mass. For example, we observed a significant decrease in SyproRuby-stained EPS with the addition of 0.5% (*w*/*w*) glucose compared to the control at 28 °C in rich media, while we observed no change in protein concentrations in the EPS in our colorimetric assay with the addition of glucose at 28 °C (Appendix A vs. Figure 3A). In minimal media, we observed no change in the amount of EPS protein when comparing added rhamnose or glucose using quantitative confocal microscopy (Appendix A).

Similarly, we examined whether the amount of carbohydrate present in the EPS changes with the addition of rhamnose for biofilms grown on agar and glass wool. We found similar trends with carbohydrate concentrations as we did with protein concentrations: in rich media for biofilms grown on agar, the added rhamnose resulted in a reduced carbohydrate concentration relative to the 0% sugar control group at 37 °C, while no change was visible with added glucose (Figure 3C). In minimal media, we observed a significant increase in carbohydrate concentration at 28 °C with rhamnose compared to the glucose control (Figure 3D). We noted no significant changes in the amount of carbohydrate in EPS as detected by calcofluor white staining and quantitation by confocal microscopy (Appendix A). Collectively, these data demonstrate that biofilm grown on rich media agar at 37 °C has less concentrated EPS protein and carbohydrate than biofilm grown without added sugar. Conversely, we noted that biofilm grown on agar in minimal media with rhamnose as the only carbon source has more concentrated protein and carbohydrate in the EPS compared to a glucose control.

### 3.3. Rhamnose Modulates Gene Expression Differently for Planktonic and Biofilm Cells

To grasp how rhamnose might be influencing the cell and modulating protein and carbohydrate concentrations in the EPS to reduce biofilm formation in rich media at warm temperatures, we curated a series of genes of interest related to biofilm formation, rhamnose transport and metabolism, and EPS production to study via quantitative polymerase chain reaction (qPCR) (Table 2).

We examined both planktonic cells and biofilm cells harvested from glass wool under all experimental conditions (Figure 4, Appendix A). In rich media, for both planktonic and biofilm cells treated with an excess of rhamnose (0.5% (*w*/*w*)), we observed increased expression of the *rhaT* transcript, which codes for the rhamnose/proton symporter that is the sole mechanism for importing rhamnose into *E. coli* (Figure 4A,B) [66]. Interestingly, in minimal media, when rhamnose is the only carbon source present, we did not see a similarly significant increase in *rhaT* compared to a 0.5% (*w*/*w*) glucose control (Figure 4C,D). We monitored the expression of two genes, *csgA* and *fimA*, that report on the extracellular structures curli and fimbriae, respectively, both of which are important for biofilm formation. We observed that *csgA* expression is increased with the addition of rhamnose in biofilm at both 28 and 37 °C, albeit only significantly in rich media at 37 °C treated with 0.5% (*w*/*w*) rhamnose; its expression in planktonic cells remained largely unchanged. For *fimA*, we observed a trend of either depressed expression (ns) or no change in expression for all experimental conditions. We also highlighted the gene *mcbA (ybiM)*, which has been demonstrated to play a role in biofilm modulation and colonic acid production and is thought to inhibit biofilm formation when overexpressed in *E. coli* [67,68]. Here, we found that, in rich media at 28 °C, 0.5% (*w*/*w*) rhamnose induced a 2.9-fold increase in *mcbA* expression (Figure 4A), whereas in minimal media supplemented with rhamnose at either growth temperature it resulted in decreased *mcbA* expression in planktonic cells (Figure 4C,D).

Given the significant differential impact of rhamnose on gene expression within our hand-picked gene set depending on experimental conditions, we were eager to gain a broader understanding of the influence of saturating concentrations of rhamnose (0.5% (*w*/*w*)) across the entire transcriptome of both planktonic and biofilm cells in rich and minimal media at either 28 °C or 37 °C. We harvested RNA from 3 biological triplicates of 16 possible experimental conditions from biofilms grown on glass wool (biofilm samples) and the surrounding culture broth (planktonic samples) and conducted RNA-seq and differential gene expression analysis on all samples. Volcano plots of differentially expressed genes for eight different conditions compared to their controls revealed that, generally, planktonic samples had the largest number of significantly differentially expressed genes, which we defined as those with a log_2_ fold change value > 2 (absolute fold change value > 4) and a false discovery rate of <0.01 (Appendix A). Biofilm samples collected at either temperature in rich media had the fewest significant differentially expressed genes, particularly compared to biofilm samples collected in minimal media. We compared those differentially expressed genes across all conditions to observe which conditions had genes that overlapped (Figure 5).

Planktonic cells had the most differentially expressed genes in general, regardless of media or temperature, and it is therefore unsurprising that we observed the greatest overlap among planktonic-to-planktonic comparisons, regardless of temperature or media type.

We were curious whether our data would cluster by media type, physiological state (biofilm vs. planktonic), or temperature. Principal component analysis (PCA) showed that across all conditions we observed clustering of samples that share media type and physiological state (Figure 6A). For example, we observed that planktonic samples grown in rich media cluster in the lower-right quadrant of the graph, while biofilm samples from most media types and temperature conditions cluster on the top half of the graph. When PCA was conducted on the data sets separated by media type, we saw clustering for rich media samples occur most dramatically among physiological type, regardless of temperature (Appendix A); we also observed considerably more spread among the biofilm replicates compared to the planktonic replicates. PCA performed on minimal media samples showed both clustering among replicates and clustering based on temperature (e.g., all 28 °C samples appear on the left half of the graph) (Appendix A).

When comparing our RNA-seq data between various experimental conditions, we observed expected and unexpected trends in gene expression modulation. When we compared our planktonic samples grown in rich media at 28 or 37 °C, we noted that transcripts from rhamnose operons *rhaBAD*, *rhaRS*, and *rhaT* were highly upregulated in response to the addition of excess rhamnose, while the *csgDEFG* operon, responsible for regulation and maintenance of curli, was downregulated at both temperatures (Figure 6B). The *carAB* operon, *pyrB*, and *upp*, which are all part of the uridine-5’-phosphate biosynthesis and salvage pathways that feed into *de novo* pyrimidine synthesis, were downregulated at 28 °C but upregulated at 37 °C in planktonic cells grown in rich media. We were not able to similarly compare biofilm samples grown in rich media at our two different experimental temperatures, as there were no overlapping significant genes. In minimal media, however, we compared biofilm grown at different temperatures and found that overlapping genes are upregulated at both temperatures (Figure 6C). As with planktonic cells in both rich and minimal media (Figure 6D), we saw the upregulation of rhamnose-related genes in biofilm cells grown at either temperature. For biofilm grown in minimal media with rhamnose, we also observed upregulation of *csgA* and *csgB* transcripts, the primary components of the external curli structure, regardless of growth temperature. In these conditions, we also observed notable upregulation of the glycolate and glyoxylate degradation pathway transcripts from the *glcDEFGBA* operon and the *prpBCDE* operon transcripts from propionate catabolism in the methyl citrate pathway. For planktonic cells grown in minimal media, we saw a strong correlation in the differential expression of overlapping genes between the 28 and 37 °C growth temperatures (Figure 6D). The *gatZABCD* operon, which transcribes enzymes in the galactitol degradation pathway, was upregulated at both temperatures, while *ibpA* and *ibpB*, which code for small heat shock proteins, were downregulated. When we compared those genes that are differentially expressed in planktonic cultures in either minimal or rich media at a given temperature, we observed some genes that are regulated similarly: transcripts from the *rhaBAD*, *rhaSRT*, and *gatZABCD* operons are all upregulated, while *mgtS*, a magnesium ion transporter, is downregulated at both 28 and 37 °C (Figure 6E and Figure 6F, respectively). We also observed many genes that were downregulated in rich media but upregulated in minimal media for planktonic cells at both temperatures; for example, many transcripts related to curli production and regulation, as well as the dipeptide ABC transporter (*dppABC* operon), followed this trend. These results highlight the global, temperature- and media-dependent impact that added rhamnose has on the bacterial transcriptome.

### 3.4. Gene Ontology Analysis Reveals Global Changes to Metabolism and Transport with Rhamnose Addition

To better understand the global impact that rhamnose addition has on the transcriptome under different growth conditions, we performed gene ontology (GO) gene enrichment analysis on our differentially expressed gene sets using PANTHER’s overrepresentation algorithm, specifically focusing on biological processes [54]. We manually grouped biological processes by their broader function to simplify the visualization of our data (Figure 7, Appendix A). We observed the greatest changes in representation in minimal media occurring in biofilm grown at 28 °C and in planktonic cells grown at 37 °C (Figure 7A, Appendix A).

For both growth temperatures in minimal media, we saw some overrepresentation of carbohydrate metabolism and transmembrane transport. At cooler growth temperatures for biofilm cells, we observed overrepresentation in nitrogen and sulfur processes and nucleobase metabolism. In rich media, we had fewer differentially expressed genes to analyze in the biofilm samples, but, nevertheless, we still observed some biological process overrepresentations in carbohydrate metabolism (regardless of temperature), as well as extracellular structure enhanced enrichment at low growth temperatures (Figure 7B, Appendix A). For planktonic cells grown in rich media, we observed notable downregulation in biosynthetic processes at warm growth temperatures, specifically carbohydrate (GO:0016051), lipid (GO:0008610), lipopolysaccharide (GO:0009103), and polysaccharide biosynthesis (GO:0000271) (Appendix A). For planktonic cells grown in rich media, we saw a consistent upregulation in amino acid metabolism, carbohydrate metabolism, and transmembrane transport.

We also used the Omics Dashboard feature in EcoCyc to examine the pathway-level changes that occur with rhamnose addition [61]. While the data analysis overlapped with our PANTHER results, we found the grouping of “Outer Membrane Proteins”, a subcategory of “Cell Exterior”, to provide noteworthy results (Figure 8). For planktonic cells in minimal media at 28 °C, we observed that many outer membrane proteins have upregulated expression, while the converse is true for planktonic cells in rich media (Figure 8A). At warmer growth temperatures, we saw upregulation of specific outer membrane protein operons (e.g., *csgBEF*) in minimal media, but otherwise downregulated expression. In planktonic cells grown in rich media at warm temperatures, we saw the upregulation of expression of *ompW*, an outer membrane protein that is involved in ion transport, and *tonB* and *exbB*, both of which are part of the Ton system for harnessing energy from the proton motive force (Figure 8B). Cells grown at warm temperatures showed upregulation and downregulation of specific outer membrane proteins, which is somewhat in contrast with cells grown at lower growth temperatures, which largely demonstrate upregulation of outer membrane proteins in minimal media and downregulation in rich media. When we analyzed our samples to observe how gene expression in the CRP regulon is changing, we noted that there is considerably more activity in the rich media samples with regards to the downregulation of considerable portions of the regulon (Appendix A). In planktonic cells grown in rich media, 56% and 58% of differentially expressed genes in the CRP regulon were downregulated at 28 °C and 37 °C, respectively (Appendix A). Conversely, in minimal media we observed largely upregulation of differentially expressed genes in the regulon, regardless of temperature or physiological state.

We also examined all of our planktonic samples, regardless of temperature and media type, in the biofilm (GO: 0042710) and adhesion (GO: 0007155) subcategories in the EcoCyc Omics Dashboard, and found that, apart from the growth conditions of minimal media at 28 °C, these factors are largely downregulated (Appendix A). Taken together with our other ontological analyses, we observed significant changes to metabolism, transport, and biofilm adhesion properties that are dependent on experimental conditions of temperature and the availability of other nutrients.

## 4. Discussion

Our results indicate L-rhamnose significantly changes growth kinetics, biofilm formation, and EPS biomolecule concentrations, and influences the transcriptome of PHL628 *E. coli*, both in planktonic and biofilm states. One of the most remarkable aspects of these data is the nuanced influence L-rhamnose has on *E. coli* gene transcription depending on the temperature of growth and the presence of other nutrients, which, to our knowledge, has never been investigated. Several groups have monitored the global transcriptomic differences that occur with *E. coli* grown on different carbon sources, including glucose, acetate, pyruvate, various amino acids, and others [69,70,71,72,73]. Smith and colleagues highlighted that genomic, metabolic, and phenotypic changes occur not only with respect to the nutrients provided, but also with respect to the lifecycle of the culture (e.g., whether they were sampling logarithmic or stationary growth phase) [73]. Our results provide an additional, comprehensive examination of the changing transcriptome based on temperature, available carbon sources, and physiological state.

Rich media (lysogeny broth) contains salt, tryptone, and yeast extract, which together contain approximately 0.16% (*w*/*w*) total carbohydrates upon preparation [31]. Other labs have explored the relationship between glucose availability in the media and the repression of *E. coli* biofilms, which is determined to be mediated by the cAMP-CRP complex, or the stimulation of growth at high glucose concentrations [33,34,35]. Phenotype microarrays have characterized the influence of various nitrogen and carbon sources on the growth and biofilm formation of clinical and agricultural *E. coli* isolates [74,75]. While these rich conditions in the laboratory are still highly defined, we assert that they mimic the more heterogeneous and complex environments that might be most vulnerable to colonization by *E. coli* biofilms, such as wastewater pipes and medical devices, compared to minimal media conditions. Therefore, cataloging how additives to these more complex conditions change biofilm formation and gene expression could lead to simple solutions to reduce biofilm formation in undesired locations.

We hone our focus on the condition of rich media at 37 °C, where we observe a significant decrease in biofilm formation with the addition of rhamnose when samples are grown on glass wool, and a reduction in protein and carbohydrate EPS concentrations. Under these conditions, we also observe interesting changes to the transcriptome of the planktonic and biofilm cells grown at this temperature. Planktonic cells grown in rich media respond to the increased concentration of rhamnose by increasing the expression of the transcription factors *rhaS* and *rhaR*, the rhamnose symporter *rhaT*, and the enzymes that initiate rhamnose catabolism found in the *rhaBAD operon*. In contrast, the expression of other genes related to carbohydrate metabolism is decreased via carbon catabolite repression. When considering differentially expressed genes in our RNA-seq data under these experimental conditions, we learn what cellular programs may promote the planktonic state. We observe significant downregulation of genes promoting adhesion and biofilm formation in planktonic cells grown at 37 °C in rich media: both the curli- and fimbriae-associated operons are downregulated, which we postulate is one of the mechanisms enhancing preference of the planktonic state with rhamnose addition. These planktonic cells also differentially express genes critical to catabolic pathways for carbohydrates—specifically rhamnose—and other biomolecules and energy processes in the cell, thus suggesting that the cells favor oxidative processes and energy production. In planktonic cells, at 37 °C we observe an absence of enriched pathways related to stress responses, which may be another mechanism causing cells to remain in their planktonic state rather than adopt a biofilm state [64,76,77,78]. We note that our results at 28 °C show little or no change in the growth of biofilm on glass wool or in its composition, as determined by confocal microscopy, with the addition of rhamnose. These results could be an artifact of the strain we are using (PHL628), which has a point mutation in the OmpR protein that leads to the overexpression of CsgD and the robust formation of curli [41,42,43]. Notably, curli form robustly at growth temperatures below 30 °C, potentially due, in part, to the selective stabilization of the Crl protein which regulates *csgD* expression [79,80]. At our warmer growth temperature (37 °C), we observe less influence of curli, which may be the reason we have more notable phenotypic changes in biofilm growth at this temperature.

The impact of rhamnose on biofilm formation and differential gene expression has broad implications for understanding bacterial adaptation and survival in different environments. With this knowledge, we could exploit the ability of rhamnose to depress biofilm formation at elevated temperatures in industrial and clinical settings so as to control biofilm-associated infections. Additionally, these insights into the influence of L-rhamnose on the transcriptome can inform new strategies for the metabolic engineering of *E. coli* for biotechnological applications. While our study provides significant perspective on the impact of rhamnose on the *E. coli* transcriptome and biofilm formation, the study primarily focuses on the short-term effects of rhamnose addition and does not explore long-term adaptation to rhamnose. Future research will investigate the long-term effects of rhamnose on bacterial physiology. Additionally, studies on the interactions between rhamnose and other environmental stressors, like antibiotics, could provide a more comprehensive understanding of its role in bacterial gene regulation and survival.

## 5. Conclusions

This study demonstrates that L-rhamnose significantly influences the growth kinetics, biofilm formation and composition, and transcriptome of PHL628 *E. coli*. Our comprehensive analysis reveals that the impact of L-rhamnose varies with temperature and media conditions. In rich media at 37 °C, L-rhamnose reduces biofilm formation and decreases the concentration of EPS carbohydrates. These changes are accompanied by the increased expression of genes involved in rhamnose transport and metabolism and the reduced expression of genes related to adhesion and biofilm formation. Our RNA-seq data further elucidate how L-rhamnose modulates the transcriptome, promoting a planktonic state and enhancing oxidative processes and energy production.

These findings broadly affect the understanding of bacterial adaptation and survival in different environments. The ability of L-rhamnose to depress biofilm formation at elevated temperatures could be exploited in clinical settings to control biofilm-associated infections and in industry to reduce biofilm-related disruptions.

## Figures and Tables

**Figure 1 microorganisms-12-01911-f001:**
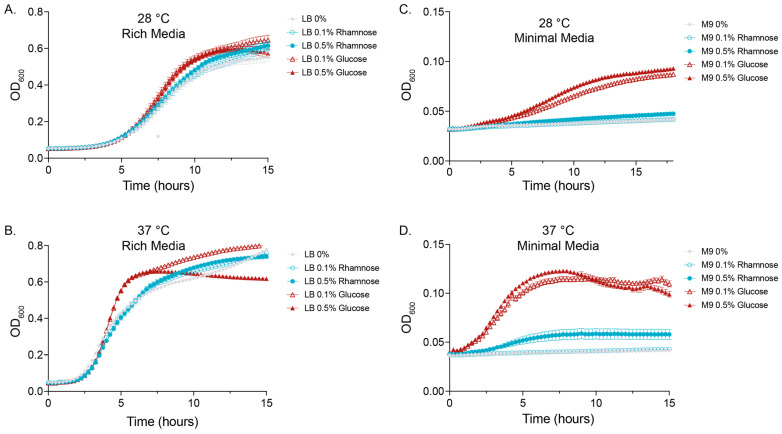
Added (0.1% or 0.5% *w*/*w*) L-rhamnose or D-glucose growth curves for PHL628 *E. coli* in (**A**,**B**) rich (LB) and (**C**,**D**) minimal (M9) media at 28 or 37 °C, as measured by OD_600_ (N ≥ 3 biological replicates). All samples grown for 15 h except for the 28 °C sample grown in M9, which required 18 h to demonstrate a growth plateau.

**Figure 2 microorganisms-12-01911-f002:**
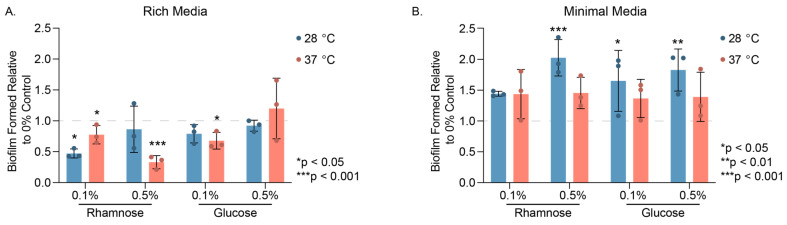
Quantitation of biofilm formed via crystal violet assay. Biofilm grew on glass wool in (**A**). rich (LB) and (**B**) minimal (M9) media conditions over 48 h at the temperatures indicated, with either 0.1% or 0.5% (*w*/*w*) L-rhamnose or D-glucose or no sugar added (0% control). Graphed data points represent biological replicates, while the magnitude of the colored bars indicate the mean and error bars are the standard deviation. Statistical significance of the sample compared to the 0% control in rich media or between 0.5% (*w*/*w*) Rha and 0.5% (*w*/*w*) Glu in minimal media was determined using two-way ANOVA analysis with a post-hoc Tukey test.

**Figure 3 microorganisms-12-01911-f003:**
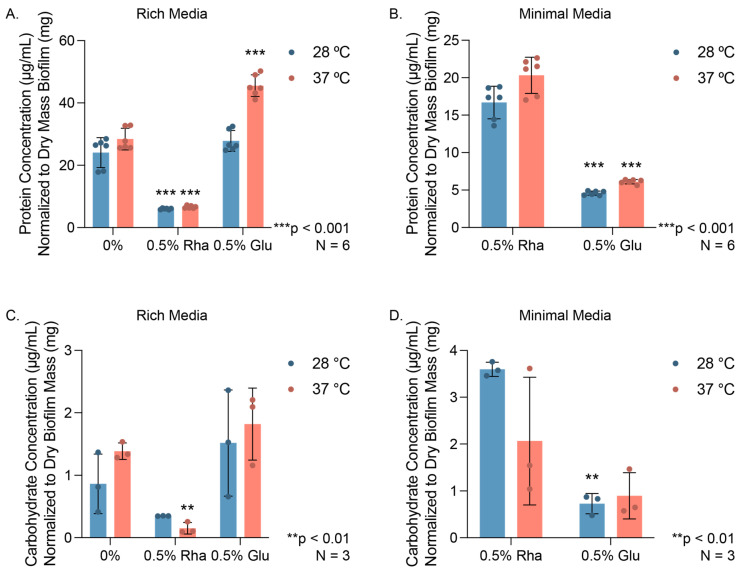
EPS biomolecule concentrations. Protein concentrations of EPS in (**A**) rich and (**B**) minimal media as determined by colorimetric bicinchoninic acid assay after 48 h growth on agar plates. Carbohydrate concentrations of EPS in (**C**) rich and (**D**) minimal media as determined by phenol sulfuric acid assay. All concentrations were normalized to the average amount of dry biofilm collected for each condition. Graphed data points represent biological replicates, while the magnitudes of the colored bars indicate the mean and error bars are the standard deviation. Statistical significance of the sample compared to the 0% control in rich media or between 0.5% (*w*/*w*) Rha and 0.5% (*w*/*w*) Glu in minimal media was determined using two-way ANOVA analysis with a post-hoc Tukey test.

**Figure 4 microorganisms-12-01911-f004:**
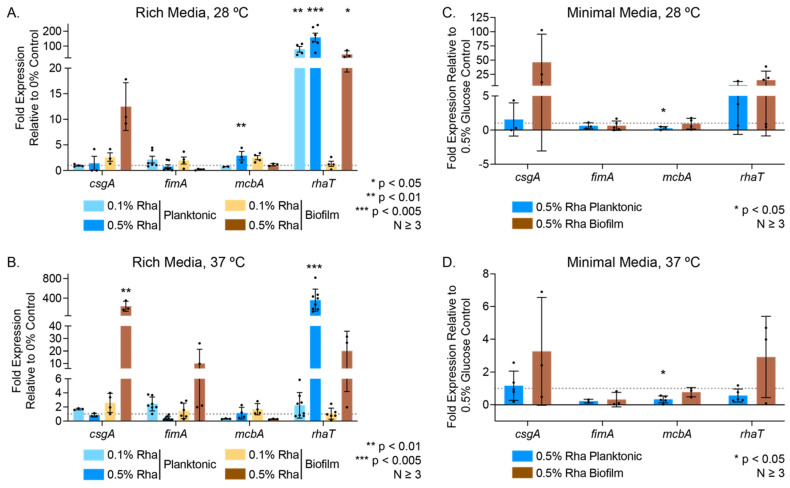
Gene expression data for select transcripts in (**A**,**B**) rich and (**C**,**D**) minimal media exposed to varying concentrations of rhamnose. Biofilms were grown on glass wool over 48 h at the temperatures indicated. Planktonic cells were harvested after 24 h of growth. The grey dashed line indicates the comparative fold expression = 1 of the control (0% sugar for LB/rich media or 0.5% (*w*/*w*) glucose for M9/minimal media). Graphed data points represent biological replicates, while the magnitude of the colored bars indicate the mean and error bars are the standard deviation. Statistical significance was determined using two-way ANOVA analysis with a post-hoc Tukey test.

**Figure 5 microorganisms-12-01911-f005:**
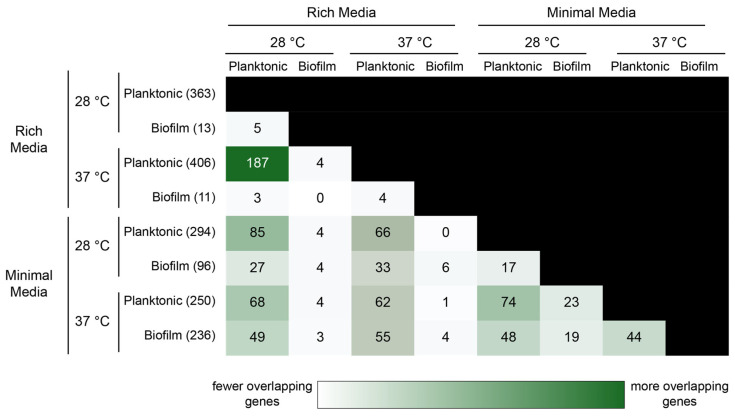
Overlapping differentially expressed genes as analyzed using edgeR. Total significant genes (log(FC) > 2, FDR < 0.01 are indicated in parentheses. Darkness of cell background indicates more overlapping genes.

**Figure 6 microorganisms-12-01911-f006:**
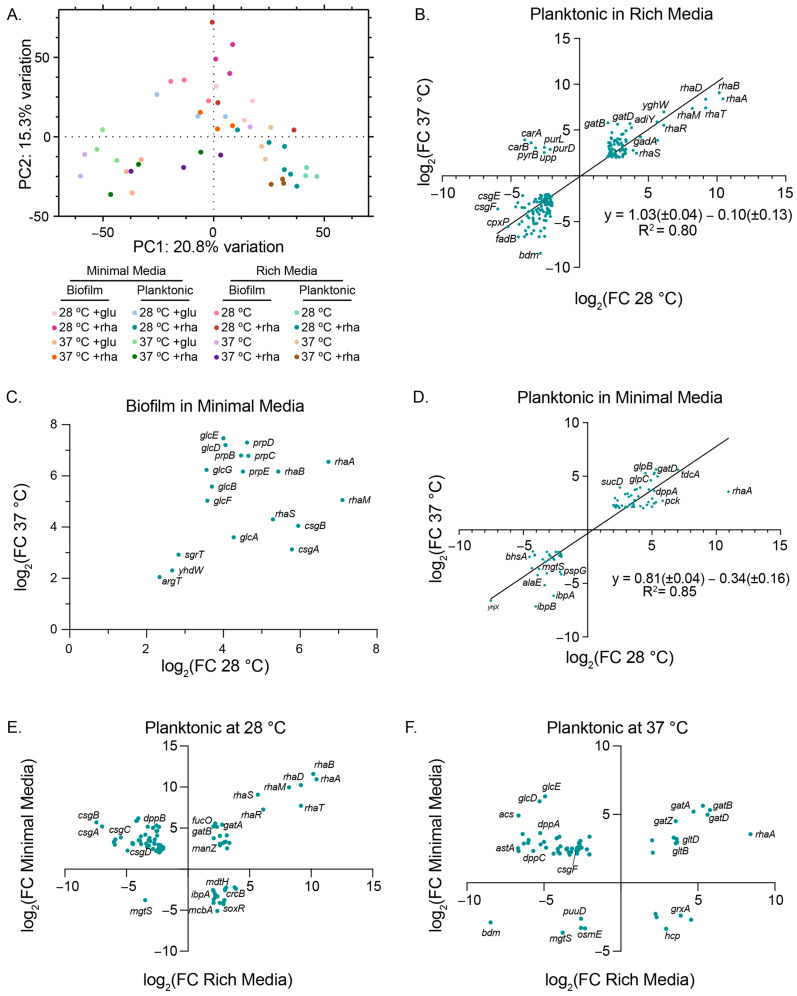
RNA-seq differential gene expression comparative analyses. (**A**) Plot of PCA scores for an analysis using all samples. Percentage displayed on axes is the rate of contribution of PC1 and PC2. Graphs of significant (log_2_(FC) > 2, FDR < 0.01) differentially expressed genes co-observed between 28 and 37 °C: (**B**) planktonic samples in rich media or (**C**) biofilm, or (**D**) planktonic samples in minimal media. Rich and minimal planktonic samples compared at (**E**) 28 °C or (**F**) 37 °C. Linear regression fits on (**B**,**D**) were completed with GraphPad Prism 10.

**Figure 7 microorganisms-12-01911-f007:**
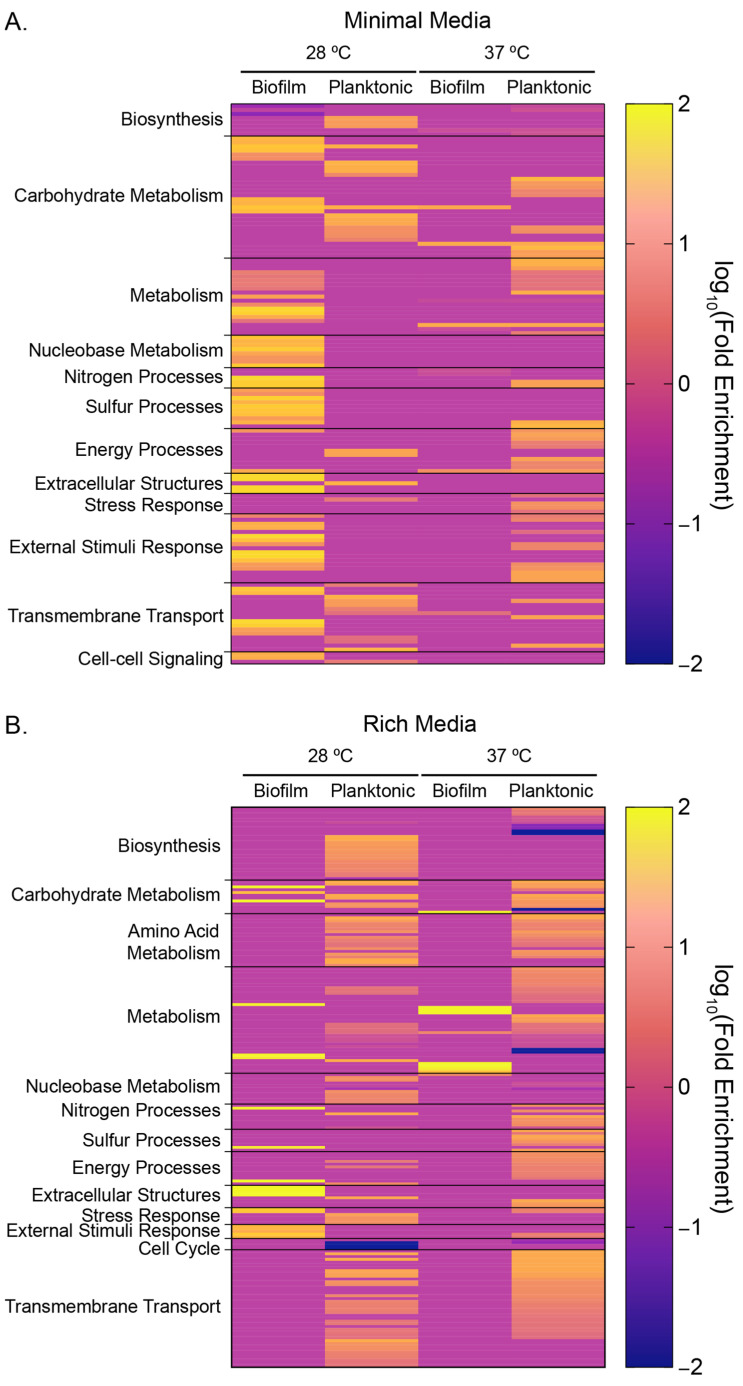
Gene Ontology Heatmaps of Overrepresented Biological Processes. PANTHER GO analysis of (**A**) minimal and (**B**) rich media for biofilm and planktonic cultures at 28 and 37 °C for differentially expressed transcripts. Broader biological categories were attributed to specific GOs, which are recorded in Appendix A.

**Figure 8 microorganisms-12-01911-f008:**
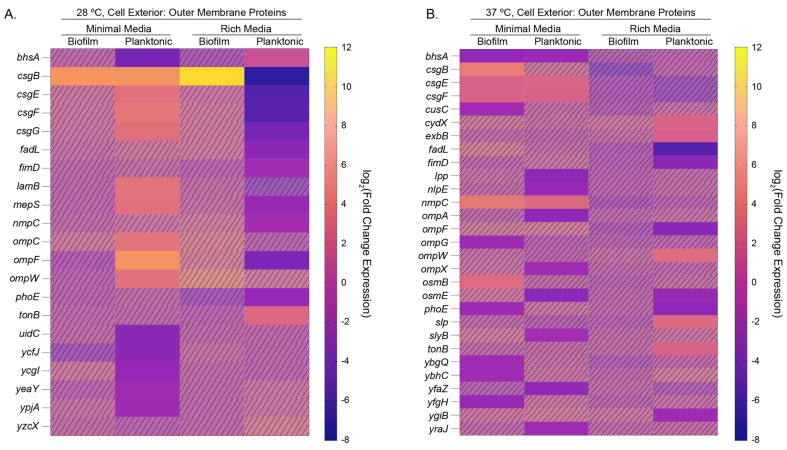
Heatmaps of the log_2_(Fold Change) expression of outer membrane proteins identified in an EcoCyc Omics analysis at (**A**) 28 °C or (**B**) 37 °C, in rich or minimal media, for biofilm and planktonic cells. Hashed and faded cells represent genes that did not have significance as determined by our edgeR differential expression analysis cutoffs of log_2_(FC) > 2 and *p* < 0.01.

**Table 1 microorganisms-12-01911-t001:** Bacterial growth data (N ≥ 3) presented in Figure 1 were fit to a logarithmic growth curve to extract the predicted rate constant (k), projected maximum growth (Y_M_), and their respective standard errors using GraphPad Prism 10. Significance was determined using a one-way ANOVA with a post-hoc Tukey test, and reported as * *p* < 0.05 comparing added sugar condition to 0%, ^§^
*p* < 0.05 comparing 0.1% (*w*/*w*) to 0.5% (*w*/*w*) within same sugar, and ^†^
*p* < 0.05 comparing rhamnose to glucose of same concentration.

		Rate Constant (hr^−1^)		Maximum Growth (Abs at 600 nm)	
		28 °C	37 °C	28 °C	37 °C
**Rich** **Media**	0%	0.469 ± 0.013	0.373 ± 0.006	0.591 ± 0.005	0.807 ± 0.004
	0.1% Rha	0.435 ± 0.012 ^†^	0.530 ± 0.008 *^,§,†^	0.635 ± 0.006 *^,§,†^	0.757 ± 0.002 *^,§,†^
	0.5% Rha	0.453 ± 0.010 ^†^	0.639 ± 0.010 *^,†^	0.659 ± 0.005 *^,†^	0.731 ± 0.002 *^,†^
	0.1% Glu	0.511 ± 0.013 *^,§^	0.873 ± 0.012 *^,§^	0.686 ± 0.005 *^,§^	0.777 ± 0.002 *^,§^
	0.5% Glu	0.69 ± 0.02 *	1.49 ± 0.03 *	0.590 ± 0.003	0.638 ± 0.002 *
**Minimal Media**	0%	0.066 ± 0.013	0.07 ± 0.02	0.047 ± 0.002	0.045 ± 0.002
	0.1% Rha	0.068 ± 0.009	0.04 ± 0.02 ^§,†^	0.048 ± 0.002 ^§,†^	0.052 ± 0.007 ^†^
	0.5% Rha	0.088 ± 0.008	0.33 ± 0.05 *^,†^	0.055 ± 0.002 *^,†^	0.059 ± 0.001 ^†^
	0.1% Glu	0.140 ± 0.005	0.66 ± 0.02 *^,§^	0.111 ± 0.003 *^,§^	0.109 ± 0.001 *
	0.5% Glu	0.185 ± 0.004	0.89 ± 0.08 *	0.1052 ± 0.0011 *	0.104 ± 0.001 *

**Table 2 microorganisms-12-01911-t002:** Genes of interest and their broad functions.

Gene Function	Genes of Interest
Biofilm formation, surface attachment	*bhsA*, *csgA*, *fimA*
EPS formation	*wcaF*, *bcsA*, *lptA*, *mcbA*
Sugar metabolism and transport	*rhaT*, *xylF*, *sfsA*, *mlc*, *crp*

## Data Availability

Workflow and source code can be accessed from the Ryno Lab Github RNA-seq Repository (https://github.com/OCRynoLab/RhamnoseRNAseq, accessed 15 September 2024). The data discussed in this publication have been deposited in NCBI’s Gene Expression Omnibus and are accessible through GEO Series accession number GSE274311 (https://www.ncbi.nlm.nih.gov/geo/query/acc.cgi?acc=GSE274311, accessed 15 September 2024).

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
