# Peer review of "L-Rhamnose Globally Changes the Transcriptome of Planktonic and Biofilm Escherichia coli Cells and Modulates Biofilm Growth"

_microorganisms, 2024, doi:10.3390/microorganisms12091911_

Round 1

Reviewer 1 Report

Comments and Suggestions for Authors

Reviewer (L-rhamnose globally changes transcriptome…. )

General comments:

                The manuscript is clearly written, detailed, and provides an extensive background on the effects of rhamnose in the introduction leading into the main body of work. The manuscript describes the detailed analysis of the effect of L-rhamnose (vs glucose) on biofilm formation (vs planktonic existence) in minimal and rich medium, low (28*C) and higher (37*C) temperature, and impact on the transcriptome by its effect on modulating gene expression. The impact of this work helps identify how L-rhamnose influences biofilm formation and (together with other such compounds) can lay the framework for methods to regulate biofilm formation on environmental or biological surfaces.

Specific comments:

1. Lines 33, 36, 37, and throughout: Sentence ending periods are set before reference brackets; they should be after the included reference.

2. Sentences should not start with a number (or abbreviation). In various places (i.e., lines 177, 197, 257, 266, elsewhere) sentences start with a number.

3. Line 252: It is not clear what method was used to remove biofilm from glass wool (was it simply the shaking method listed in lines 255-259?). Shouldn’t biofilm attach firmly and resist detachment by shaking except for loosely bound cells? And those that did detach by shaking be almost considered ‘planktonic’ based on the availability of nutrients at the outer periphery of a biofilm?

4. Figures 2, 3, and 4.  Please identify in the legend the significance of the colored dots that indicate temperature (28* vs 37*C) when lined up along the error bar stem, when on either side, or sitting at the top or bottom hash mark.

5. Figure 5. Could the images that make up Fig. 5 be larger(?) as they are barely discernable.

Author Response

We thank the reviewer for their favorable description of our paper and for their time in completing this review.

Specific comments:

  1. Lines 33, 36, 37, and throughout: Sentence ending periods are set before reference brackets; they should be after the included reference.

We thank the reviewer for noticing this editing error and have revised the manuscript accordingly.

  1. Sentences should not start with a number (or abbreviation). In various places (i.e., lines 177, 197, 257, 266, elsewhere) sentences start with a number.

We thank the reviewer for noticing this editing error and have revised the manuscript accordingly.

  1. Line 252: It is not clear what method was used to remove biofilm from glass wool (was it simply the shaking method listed in lines 255-259?). Shouldn’t biofilm attach firmly and resist detachment by shaking except for loosely bound cells? And those that did detach by shaking be almost considered ‘planktonic’ based on the availability of nutrients at the outer periphery of a biofilm?

We agree with the reviewer that biofilm is usually resistant to shaking and should remain attached to the glass wool. We observe that loosely bound cells would be rinsed away in the three buffer rinses prior to shaking and dislodging of the biofilm with glass beads. We use glass beads to mechanically disturb the biofilm and detach the cells. We adopted this method from reference Benamara et al. 2011, reference 45, which we have now also included in our methods section (Line 147-148).

  1. Figures 2, 3, and 4.  Please identify in the legend the significance of the colored dots that indicate temperature (28* vs 37*C) when lined up along the error bar stem, when on either side, or sitting at the top or bottom hash mark.

We thank the reviewer for helping us clarify our figures. We now indicate in the legends for these figures that each circle is a biological replicate, with the magnitude of the bar representing the mean and the error bar representing the standard deviation (Lines 389, 432, 498). The alignment of the individual was a setting in our data visualization program (GraphPad Prism) and has no significance. We believe that the program puts dots that are similar in value horizontally so they can be visualized and do not overlap.

  1. Figure 5. Could the images that make up Fig. 5 be larger(?) as they are barely discernable.

We thank the reviewer for noticing this lack of clarity. We apologize for this oversight and have made Figure 5 larger to take up a full page (p 15) and enlarged some of the in-figure text, too.

Reviewer 2 Report

Comments and Suggestions for Authors

This manuscript has been reported that L-rhamnose widely influences cellular metabolism of E. coli resulting in changing kinetics of cell growth and biofilm formation. To understand the impact of rhamnose addition on E. coli K-12, they compared kinetics of growth, production of protein and polysaccharides, and transcriptome among four culture states: planktonic cells and biofilms at 28 °C and 37 °C.  As described by the authors, understanding the effect of rhamnose addition to E. coli culture on biofilm formation gave important information about regulating and inhibiting pathogenic bacteria biofilms, and the results of transcriptomic analysis provided the knowledge of the main metabolic pathway(s).

Unfortunately, this manuscript has some mistakes in grammar, less explanation the result of RNA-seq and inadequate discussion in transcriptome analysis. I strongly recommend this should be revised for the readers understanding clearly the worthy of accomplishments by the authors.

1. Mixture of present and past tenses

L.452-458, 460-469, 521-554, 589-601, 610-628, 666-683

These sentences are present tense.

In explanation about results, usually the sentences are expressed in past tense.

Please check all sentences correct or error in grammar again.

I recommend you should learn about scientific writing such a following website:

https://www.matrix.edu.au/how-to-write-a-scientific-report/.

2. Expression of gene names

In this manuscript, several relevant genes’ names were combined (e.g. rhaA rhaB and rhaD rhaBAD). But this style is not correct. Therefore, you should replace such ‘unique original combined names” to correct gene names.

3. About Figure 2 and 3

In these figures, the statistical results (differences) are shown as *, **, and ***. What are control conditions? Please add the information.

4. Gene oncology analysis

The authors used GO numbers instead of gene names, compared gene expressions’ level value as in the metabolic pathways. However, there is no information about which GO number indicate gene name. Could you add this information in main text or supplemental data?

5. database registration of RNA-seq data

Have you register the original RNA-seq data to database such as NCBI? Do you gain the SRA number? If you have, please write it down.

I recommend you should register your data (sets).

Reference: https://www.amnh.org/research/staff-directory/robert-desalle/rna-seq-data-submission#:~:text=Go%20to%20the%20Sequence%20Read,the%20top%20of%20the%20page).

6. Visualizing the metabolic pathway

To understand key factors (genes, operons) influenced by rhamnose addition more easily, could you describe the predictive cellular pathway(s) cascades using gene names?

Comments on the Quality of English Language

You should check all text again correct or incorrect in grammar before you resubmit the revised manuscript.

Author Response

We thank the reviewer for their time in reading and providing feedback on this manuscript. We have added more depth to the discussion of our RNA-seq results which we hope satisfies the reviewer.

  1. Mixture of present and past tenses

L.452-458, 460-469, 521-554, 589-601, 610-628, 666-683

These sentences are present tense.

In explanation about results, usually the sentences are expressed in past tense.

Please check all sentences correct or error in grammar again.

I recommend you should learn about scientific writing such a following website:

https://www.matrix.edu.au/how-to-write-a-scientific-report/.

We thank the reviewer for noting our oversight of using a mixture of tenses. We have combed through the manuscript to maintain past tense throughout our reporting of methods and results. We have also used the software Grammarly to confirm some of our grammar choices throughout the manuscript. We are happy to work with the MDPI copy editors to manage any further minor grammar changes specific to the journal.

  1. Expression of gene names

In this manuscript, several relevant genes’ names were combined (e.g. rhaA rhaB and rhaD àrhaBAD). But this style is not correct. Therefore, you should replace such ‘unique original combined names” to correct gene names.

We thank the reviewer for carefully reading through our manuscript. It is a common practice to combine genes within the same operon in E. coli, which we articulate when combining them in this fashion. We overlooked this articulation in the example the reviewer notes (on line 54) which we have now corrected. The reviewer can become more familiar with this practice by considering the following manuscript, which references the commonly used lac operon in E. coli:

Osbourn, A. E., & Field, B. (2009). Operons. Cellular and molecular life sciences : CMLS, 66(23), 3755–3775. https://doi.org/10.1007/s00018-009-0114-3

  1. About Figure 2 and 3

In these figures, the statistical results (differences) are shown as *, **, and ***. What are control conditions? Please add the information.

We thank the reviewer for highlighting this oversight in clarity. The asterisks in these figures indicate the significance between the sample and control (for LB/rich media, this would be 0%; for minimal media, the asterisks denote a difference between the 0.5% rhamnose and 0.5% glucose samples). We have indicated this in the figure legends now using this sentence: “Statistical significance of the sample compared to the 0% control in rich media or between 0.5% Rha and 0.5% Glu in minimal media was determined using two-way ANOVA analysis with a post-hoc Tukey test.” on lines 387 and 432.

  1. Gene oncology analysis

The authors used GO numbers instead of gene names, compared gene expressions’ level value as in the metabolic pathways. However, there is no information about which GO number indicate gene name. Could you add this information in main text or supplemental data?

GO numbers do not reflect a specific gene name, but rather a pathway that a collection of genes that are, in our case, overrepresented for these experimental conditions. Supporting Tables S3 and S4 relay the GO numbers associated with the broad categories described in Figure 6. In the main text, we acknowledge the following GOs:

carbohydrate biosynthetic processes (GO:0016051)

lipid biosynthetic process (GO:0008610)

lipopolysaccharide biosynthetic process (GO:0009103)

polysaccharide biosynthetic process (GO:0000271)

in addition to the most interesting GO categories, GO: 0042710 and GO: 007155 (biofilm and adhesion, respectively), which we highlighted and presented this gene-level data in Supporting Figure S7. We have now added data sheet tabs to Supporting Table S4 that highlight these GO categories at the gene level in rich media (genes were found using Ecocyc) and report the differential gene expression, per this reviewer’s request.

Additionally, readers will be able to use these category identifiers for particular pathways that they are interested in and our processed differential gene expression data in Supporting Table S2.

  1. database registration of RNA-seq data

Have you register the original RNA-seq data to database such as NCBI? Do you gain the SRA number? If you have, please write it down.

I recommend you should register your data (sets).

Reference: https://www.amnh.org/research/staff-directory/robert-desalle/rna-seq-data-submission#:~:text=Go%20to%20the%20Sequence%20Read,the%20top%20of%20the%20page.

We thank the reviewer for their suggestion. We have deposited our data at the NCBI, as reflected in Line 301-303: “The data discussed in this publication have been deposited in NCBI's Gene Expression Omnibus[44] and are accessible through GEO Series accession number GSE274311 (https://www.ncbi.nlm.nih.gov/geo/query/acc.cgi?acc= GSE274311).” It is currently embargoed until September 30th, 2024 or until the acceptance of this manuscript.

  1. Visualizing the metabolic pathway

To understand key factors (genes, operons) influenced by rhamnose addition more easily, could you describe the predictive cellular pathway(s) cascades using gene names?

We thank the reviewer for their question. Using gene names to describe pathways would be extremely text-dense and, in our opinion, not as effective as using the gene ontology grouping categories we reported in Figure 6. We report the specific GO numbers and finer-grained details about our enrichment values in Supporting Tables S3 and S4, including some newly added data in response to question 4 above.

Reviewer 3 Report

Comments and Suggestions for Authors

The manuscript submitted by Hantus et al. described that the L-rhamnose globally changes the transcriptome of planktonic and biofilm Escherichia coli cells and reduces biofilm growth at 37 °C. The study is important and interesting. The following are some suggestions to revise the manuscript:

Introduction:

1. The reason why the author used L-rhamnose for E. coli biofilm experiments is not clarified. The introduction should include relevant studies on the effects of rhamnose on E. coli biofilms or other bacterial biofilms.

2. Rhamnose is the primary carbohydrate component of rhamnolipids, and do they have the same effect on biofilms?

Materials and Methods:

1. The methods section lacks relevant references.

2. In 2.3 crystal violet assay section, why the biofilm of E. coli was measured at 48 hours.

3. The subtitle of “Confocal.” should be revised.

3. The specific company responsible for sequencing should be named in 2.7 RNA-seq data collection and analysis section.

4. Authors need to check and confirm the name of RT-PCR. RT-PCR or qPCR?

5. There should be a brief description of housekeeping gene “rrsA”.

Results:

1. In fig 1C, the growth of E. coli is not completed. What is the maximal biomass of the growth in minimal media at 27 °C? in my opinion, the data of figure 1 and table 1 was repetitive.

2. In table 2, the authors listed 12 genes, why did RT-PCR validate only four of them.

3. The error bar of the data in Fig 3 and Fig 4 is too high with lower reliability.

4. Authors should use RT-PCR to verify the accuracy of RNA-seq data.

Discussion:

1. L-rhamnose did not significantly reduce E. coli biofilm formation at 28 °C, the reasons for this result should be discussed in detail.

2. The results of confocal experiments should be discussed.

Comments on the Quality of English Language

The writing of English should be revised with minor modifications. for example, The writing of “pH 7.0 0.2 M potassium phosphate” should be changed; “Tris-HCl” not “Tris-HCL” or “ Tris-Cl”.

Author Response

Introduction:

  1. The reason why the author used L-rhamnose for E. colibiofilm experiments is not clarified. The introduction should include relevant studies on the effects of rhamnose on E. colibiofilms or other bacterial biofilms.

We apologize for this oversight and thank the reviewer for bringing this to our attention. In addition to what we have included in the introduction about rhamnolipids (see Q2 below), which appear to be much more extensively studied than L-rhamnose, we also have identified two papers that look at rhamnose and other organisms. We also cite literature that looks at the impact of other sugars on E. coli biofilms (though these studies do not include rhamnose) (Lines 79-84).

“Other labs have explored the relationship between glucose availability in the media and repression of E. coli biofilms, and have explored the phenotypic effects of some sugars, but not rhamnose, on E. coli [33–38]. The effect of rhamnose on biofilm formation and composition has been documented for other gram-negative organisms like Flavobacterium columnare and Phaeobacter inhibens, but, to our knowledge, not in E. coli [39,40].”  

  1. Rhamnose is the primary carbohydrate component of rhamnolipids, and do they have the same effect on biofilms?

We thank the reviewer for pointing this idea out and for their question. Rhamnolipids do indeed have interesting effects on biofilms, including biofilm dispersal. We did not elaborate on them beyond the text below, as they are most associated with Pseudomonas aeruginosa biofilms rather than E. coli biofilms. We include in the introduction the following text (Current lines 34-38):

“Rhamnose is also the primary carbohydrate component of rhamnolipids: glycolipid biosurfactants that are predominantly produced by Pseudomonas aeruginosa and are notable in their ability to modulate biofilm formation, maturation, and dispersal [6–9]. Notably, groups have developed rhamnolipid mimics as a novel class of anti-biofilm compounds [10,11].”

Materials and Methods:

  1. The methods section lacks relevant references.

We thank the reviewer for their comment and apologize for the oversight. In addition to the references already included in the methods, we have added the following:

Biofilm growth on glass wool (Line 148, Ref 45): Benamara, H.; Rihouey, C.; Jouenne, T.; Alexandre, S. Impact of the Biofilm Mode of Growth on the Inner Membrane Phospholipid Composition and Lipid Domains in Pseudomonas Aeruginosa. Biochimica Et Biophysica Acta Bba - Biomembr 2011, 1808, 98–105, doi:10.1016/j.bbamem.2010.09.004.

Crystal violet microtiter assay (Line 159, Ref 46): O’Toole, G.A. Microtiter Dish Biofilm Formation Assay. J Vis Exp Jove 2011, doi:10.3791/2437.

Biofilm growth on agar plates (Line 179, Ref 47): Chiba, A.; Sugimoto, S.; Sato, F.; Hori, S.; Mizunoe, Y. A Refined Technique for Extraction of Extracellular Matrices from Bacterial Biofilms and Its Applicability. Microb Biotechnol 2015, 8, 392–403, doi:10.1111/1751-7915.12155.

Calcofluor white staining was done as directed by Sigma-Aldrich (the manufacturer), which is now made clear in Line 229.

  1. In 2.3 crystal violet assay section, why the biofilm of E. coli was measured at 48 hours.

We thank the reviewer for their question. We find that there are suitable, reproducible levels of biofilm growth on glass wool after 48 h. Other experiments in the lab (Buck, et al. 2021) looked a different biofilm growth timing, but found that 48 h achieved consistent results.

  1. The subtitle of “Confocal.” should be revised.

We thank the reviewer for noticing this error. We apologize for the oversight and have corrected the subtitle to “Confocal Microscopy”.

  1. The specific company responsible for sequencing should be named in 2.7 RNA-seq data collection and analysis section.

The specific company (Mr. DNA, Molecualr Research LLP) is mentioned in the fourth sentence of sub-section 2.7 in the methods section, Line 297: “Samples were analyzed by Mr.DNA (Molecular Research LP, Shallowater, Texas, USA).

  1. Authors need to check and confirm the name of RT-PCR. RT-PCR or qPCR?

We thank the reviewer for alerting us to this need for clarification. RT-PCR is reverse transcription PCR, which is conducted on the RNA transcripts harvested from our samples to convert RNA to cDNA, which is more robust. This cDNA can be used for quantitation (qPCR) using fluorescence-based detection methods (SYBR green intercalates double stranded DNA) coupled to PCR amplification of select DNA as guided by primers specific to genes of interest. We attempted to clarify our use of these terms throughout, especially in the methods section (Lines 338-339): “Prior to quantification of samples through qPCR, purified RNA must be converted into cDNA via reverse transcription (RT). RT-PCR reactions were performed to prepare cDNA from total cellular RNA using the QuantiTect Reverse Transcription Kit (Qiagen).”

  1. There should be a brief description of housekeeping gene “rrsA”.

We thank the reviewer for this suggestion and agree with them. We have added the following text to the methods section (Line 343-344, Ref 57): “All transcripts were normalized to the housekeeping gene rrsA, which was selected due to other labs’ validation of the gene [57], and all measurements were performed in triplicate.”

Ref 57: Zhou, K., Zhou, L., Lim, Q. ’En, Zou, R., Stephanopoulos, G., and Too, H.-P. (2011) Novel reference genes for quantifying transcriptional responses of Escherichia coli to protein overexpression by quantitative PCR. BMC Mol. Biol. 12, 18–18

Results:

  1. In fig 1C, the growth of E. coli is not completed. What is the maximal biomass of the growth in minimal media at 27 °C? in my opinion, the data of figure 1 and table 1 was repetitive.

We thank the reviewer for noticing this in Figure 1C. We intended to keep the x-axis scale identical across all experimental conditions, but upon the reviewer’s suggestion, we extended the x-axis for Figure 1C to be out to 18 h and demonstrate plateauing of growth. We agree that the information in Table 1 and Figure 1 are identical and represented differently. We believe the data in Table 1 is important enough to keep in the main text, as it highlights the quantitative differences in the growth rates and maximal growth achieved for the different experimental conditions. The biomass growth is reported in Supporting Figure S1.

  1. In table 2, the authors listed 12 genes, why did RT-PCR validate only four of them.

We thank the reviewer for their comment. The data for the other eight genes are validated and probed by qPCR is available in the Supplemental Information (Figure S3).

  1. The error bar of the data in Fig 3 and Fig 4 is too high with lower reliability.

We agree that we see large errors for some of our genes and experimental conditions; we note that we do not observe statistical significance for these genes. In many cases, we attempted to obtain additional biological replicates to include in our analyses with the hope that we could either remove an outlier using Grubb’s analysis or provide additional statistical power for our analysis. We do not report any significance associated with data that has high error. Importantly, this can reflect the challenging nature of the experiments and the inherent variability of the technique.

  1. Authors should use RT-PCR to verify the accuracy of RNA-seq data.

We appreciate this suggestion from the reviewer. We have indeed included qPCR data (Figure 4 and Supporting Figure S3) that corroborate our RNA-seq data (we just happened to present it before the RNA-seq data for the sake of narrative flow of our results). While RNA-seq validation was a common practice in the 00’s and early 10’s, the accuracy of modern RNA-seq data, especially when compared to microarray data collection, is such that with biological replicate analysis, there is little need to validate results. See this excellent paper, which demonstrates that there are certain types of transcripts (e.g., short) that lend themselves to problematic RNA-seq results, but that the majority (>85%) of the transcripts mapped using our workflow (Salmon, in particular) are in agreement: Everaert, C., Luypaert, M., Maag, J. L. V., Cheng, Q. X., Dinger, M. E., Hellemans, J., & Mestdagh, P. (2017). Benchmarking of RNA-sequencing analysis workflows using whole-transcriptome RT-qPCR expression data. Scientific reports, 7(1), 1559. https://doi.org/10.1038/s41598-017-01617-3. There is also a useful opinion piece on this matter that argues that additional validation, especially when done by cherry-picking relevant genes (which is what most do), is not necessary: Coenye T. (2021). Do results obtained with RNA-sequencing require independent verification?. Biofilm, 3, 100043. https://doi.org/10.1016/j.bioflm.2021.100043.

Discussion:

  1. L-rhamnose did not significantly reduce E. colibiofilm formation at 28 °C, the reasons for this result should be discussed in detail.

We thank the reviewer for this suggestion. We believe that the lower growth temperature has the added advantage of enhanced curli (beyond what we would see with our overexpression system), which may be muting any changes that rhamnose might have on biofilm formation. At warmer growth temperatures, curli does not form as favorably, which may be why the rhamnose-induced transcriptomic changes lead to phenotypic changes in biofilm formation: we are not seeing an overwhelming and potentially overshadowing impact of curli at 37 °C. We include this idea in our discussion on lines 722-730, quoted below:

“We note that our results at 28 °C, which show little or no change in growth of biofilm on glass wool or its composition, as determined by confocal microscopy, with the addition of rhamnose. These results could be an artifact of the strain we are using (PHL628), which has a point mutation in the OmpR protein that leads to overexpression of CsgD and robust formation of curli [41–43]. Notably, curli form robustly at growth temperatures below 30 °C, potentially due in part to the selective stabilization of the Crl protein which regulates csgD expression [73,74]. At our warmer growth temperatures (37 °C) , we observe less influence of curli, which may be why we have more notable phenotypic changes in biofilm growth at this temperature.”

  1. The results of confocal experiments should be discussed.

We thank the reviewer for suggesting this. Our confocal data was collected only at 28 °C, and only showed significant depression of protein in the EPS (via SYPRO-Ruby staining) when 0.5% glucose was added. We note the lack of significance in the text in the Results section (Lines 441 – 447 and 455-457) and highlight the importance of biofilm growth substrate. At the reviewers’ request have also briefly mentioned our confocal results in the discussion in reference to our lack of biofilm growth change at 28 °C with rhamnose, which is also reflected by our confocal experiments (Line 722-724):

“We note that our results at 28 °C, which show little or no change in growth of biofilm on glass wool or its composition, as determined by confocal microscopy, with the addition of rhamnose.”

Round 2

Reviewer 2 Report

Comments and Suggestions for Authors

The presentation has been developed enough to accept the manuscript, but some sentences are still necessary to check English grammar before acceptance.

Comments on the Quality of English Language

Please use an English-editing service before publication.

Author Response

We thank the reviewer for their comments.